Forest structure dependency analysis of L-band SAR backscatter

Ji Yongjie 1
Huang Jimao 1
Ju Yilin 2
Guo Shipeng 1
Yue Cairong cr_yue@126.com 2
1 Southwest Forestry University, School of Geography and Ecotourism , Kunming , Yunnan , China
2 Southwest Forestry University, Forestry College , Kunming , Yunnan , China
Yu Le
Electronic publication date: 2020 Sep 30
Publication date: 2020
Volume: 8
Electronic Location ID: e10055
Received 2020 Jul 14; Accepted 2020 Sep 7
Copyright: ©2020 Ji et al.
Copyright year: 2020
Copyright holder: Ji et al.
License: This is an open access article distributed under the terms of the Creative Commons Attribution License, which permits unrestricted use, distribution, reproduction and adaptation in any medium and for any purpose provided that it is properly attributed. For attribution, the original author(s), title, publication source (PeerJ) and either DOI or URL of the article must be cited.
License URL: https://creativecommons.org/licenses/by/4.0/

Keywords: Forest structure, L-band, SAR, Backscatter

Funding: The National Key R&D Program of China 2017YFB0502700 The National Natural Science Foundation of China 31860240 The Scientific Research Fund of Yunnan Education Department “Study on the method of improving saturation point by remote sensing of forest biomass – Taking multi-polarization SAR technology as an example” 2019J0182 The Scientific research fund of Yunnan Education Department ”Forest biomass retrieval based on GF-3 SAR data” 2020Y0393 This research was funded by the National Key R&D Program of China (2017YFB0502700), the “National Natural Science Foundation of China (No. 31860240)”, the Scientific Research Fund of Yunnan Education Department “Study on the method of improving saturation point by remote sensing of forest biomass – Taking multi-polarization SAR technology as an example (No. 2019J0182)” and by the Scientific Research Fund of Yunnan Education Department “Forest biomass retrieval based on GF-3 SAR data (No. 2020Y0393)”. The funders had no role in study design, data collection and analysis, decision to publish, or preparation of the manuscript.

==============================
Forest structure plays an important role in forest biomass inversion using synthetic aperture radar (SAR) backscatter. Synthetic aperture radar (SAR) sensors with long-wavelength have the potentiality to provide reliable and timely forest biomass inversion for their ability of deep penetration into the forest. L-band SAR backscatter shows useful for forest above-ground biomass (AGB) estimation. However, the way that forest structure mediating the biomass-backscatter affects the improvement of the related biomass estimation accuracy. In this paper, we have investigated L-band SAR backscatter sensitivity to forests with different mean canopy density, mean tree height and mean DBH (diameter at breast height) at the sub-compartment level. The forest species effects on their relationship were also considered in this study. The linear correlation coefficient R, non-linear correlation parameter, Maximal Information Coefficient (MIC), and the determination coefficient R2 from linear function, Logarithmic function and Quadratic function were used in this study to analyze forest structural properties effects on L-band SAR backscatter. The HV channel, which is more sensitive than HH to forest structure parameters, was chosen as the representative of SAR backscatter. 6037 sub-compartment were involved in the analysis. Canopy density showed a great influence on L-band backscatter than mean forest height and DBH. All of the R between canopy density and L-band backscatter were greater than 0.7 during the forest growth cycle. The sensitivity of L-band backscatter to mean forest height depends on forest canopy density. When canopy density was lower than 0.4, R values between mean forest height are smaller than 0.5. In contrast, the values of R were greater than 0.8 if canopy density was higher than 0.4. The sensitivity SAR backscatter to DBH fluctuated with canopy density, but it only showed obvious sensitivity when canopy density equals to 0.6, where both the linear and non-liner correlation values are higher than others. However, their effects on L-bang HV backscatter are affected by forest species, the effects on three forest structural parameters depend on tree species.

Introduction

Forest structure plays an important role in forest biomass inversion using synthetic aperture radar (SAR) backscatter. Forest biomass inversion using SAR data has been one of the hot topics since the early 1990s (Toan et al., 2001). Total forest biomass includes both aboveground biomass (AGB; e.g., trees, shrubs, and vines) and belowground biomass (e.g., living roots, dead fine and coarse litter associated with soil). Due to the penetration limitation of SAR and difficulty in collecting field survey data of belowground biomass, AGB usually works as forest biomass in the majority of biomass studies (Lu et al., 2014). Previous studies confirmed the potentiality of SAR backscatter in forest biomass inversion using different microwave bands, the involved bands included C-band (3.8 cm∼7.5 cm), L-band (15.0 cm∼30.0 cm) and P-band (30.0 cm∼100 cm). Meanwhile, they suggested forest structure dependency of the SAR backscatter, which reduced the sensitivity of SAR backscatters to forest biomass and its saturation level (Santoro et al., 2012; Peregon & Yamagata, 2013). SAR backscatter does not provide a direct measurement of forest biomass, it is determined by a variety of forest structure parameters that may, or may not, correlate with AGB. If the forest structure is well-correlated with AGB, the SAR backscatter will be so, however, if the forest structure is not the same trend to AGB, it is not the case for SAR backscatter. Moreover, the forest of similar biomass but different structural could be exploited to separate as dissimilar forest types (Imhoff, 1995; Woodhouse et al., 2012). Therefore, to better understand and interpret the biomass-SAR backscatter relationship, especially in real physical terms, it is important to understand forest structure effects on SAR backscatter.

L-band and P-band were demonstrated more sensitive to forest biomass estimation than C-band and X-band because they have the longer wavelength, which has a stronger capability to penetrate forest canopy capturing more backscattering from branch and trunk. As the shortage of P-band space-borne data, most SAR backscatter-based biomass estimation studies use L-band data (Lu et al., 2014). Reports from Advanced Land Observing Satellite Kyoto & Carbon also demonstrated great potential of L-band SAR data to provide accurate estimates in global forest biomass, according to their work of Phase 1,2,3 and Phase 4 currently (Santoro et al., 2012; Pantze, Santoro & Fransson, 2014; Fransson et al., 2016; Stelmaszczuk-Górska et al., 2018). In recent years, large L-band data sets were collected from Advanced Land Observing Satellite Phased Array type L-band Synthetic Aperture Radar (ALOS-1 PALSAR-1) and its successor ALOS-2 PALSAR-2, which has higher spatial resolution and shorter satellite revisit time. The new satellite missions, like NISAR (NASA-ISRO Synthetic Aperture Radar) and TanDEM-L, would have provided L-band SAR data, can be launched in 2020 and 2022, respectively. These emerging data requires a deep understanding in how forest structure mediates the relationship of biomass and L-band SAR backscatter. Without fully understanding of the interaction between forest structure and L-band SAR backscatter, forest biomass estimation with L-band backscatter runs the risk of low accuracy and low saturation level.

To address this challenge, previous studies carried out with various ways and different forest structure biophysical parameters. The most accurate way of exploring forest structure effect on SAR backscatter is relying on forest microwave scattering models, developed from the radiative transfer equation and an individual tree model. Michigan Microwave Canopy Scattering Model (MIMICS) was proposed and applied in forest backscattering analysis at X-, C- and L-band. Backscattering from the total crown, ground-trunk, and the direct ground was distinguished successfully with this model for different forest (Mcdonald, Dobson & Ulaby, 1990; Ulaby et al., 1990; Wang & Day, 1993). Results from previous studies were important for us to understand forest scattering mechanisms and explain the response of SAR backscatter from different components of forest biomass, but they are unsuitable and not enough for understanding forest structure mediation for biomass- SAR backscatter relationship. This is because the forest structure parameters described in MIMICS like forest crown and trunk are inappropriate to depict the accumulation of forest biomass. Instead of the mentioned structure parameters, effects analysis of trunk height, trunk diameter, stem density, and leaf area index (LAI), which change with the accumulation of biomass, is more useful and appropriate for over landscapes biomass estimation.

While the significance of biomass accumulation related forest structure parameters analysis, their difficulties in defining and measuring differences in plant morphology at a sub-compartment level and in site selecting that exhibit strong structural difference pose a challenge for dealing with their potential impact on SAR backscatter (Mcdonald, Dobson & Ulaby, 1990). Since the shortage of SAR and in-situ data, early studies conducted experiments based on links between MIMICS and forest growth models. For example, Kassichke et al. discovered the effects of trunk height, trunk diameter, canopy density, and LAI on SAR backscatter at X-, C- and L-band (Kasischke & Christensenjr, 1990; Karam et al., 1995). Imhoff (1995) extended the study of Kasischke into stand level with a parameter named vegetation surface-area-to-volume ratio (SA/V) and demonstrated the substantial effect of structural variation at C-, and P-band backscatter. It concluded that a universal SAR biomass equation for Earth’s forest is impossible and suggested more research should be carried out using data sets designed to test forest structure impacts (Imhoff, 1995). Among all of the biomass accumulation related forest structure parameters, individual tree height or forest height at a stand or a sub-compartment level attracted more interests for the developed interferometric SAR (InSAR), polarimetric SAR interferometry (PolInSAR), tomographic SAR techniques and their better performance for forest height-related biomass estimation. However, forest height estimation accuracy is affected by forest structure either, especially by forest canopy density (Treuhaft & Siqueira, 2000; Cloude & Papathanassiou, 2003; Cloude, 2006; Balzter, Rowland & Saich, 2007; Kugler et al., 2014; Zhu et al., 2018). Studies of forest biomass inversion with SAR techniques nowadays presented or confirmed influence of biomass accumulation related forest structure parameters, but, to our best knowledge, none of them gave a feasible way to analyze their effects or give an explicit interpretation of how these parameters impact SAR backscatter (Cartus, 2010; Golshani, Maghsoudi & Sohrabi, 2019).

A large data set of images and in-situ data from national forest inventory available over global landscapes enable investigation of forest structure effects in large-scale regions in this decade. Several efforts performed recently devoted to finding a feasible way to give an explicit explanation of forest structure effects on SAR backscatter. Manabu et al. (2006) explored forest species dependence of L-band ALOS-1 PALSAR-1 data and found dependency was absent at VV channel and low at HV channel. Santoro et al. (2009) discovered HV backscatter presented stronger sensitivity to forest growth stage than other channels. Biomass accumulation related forest structure parameters like mean forest height and DBH were explored in several works of literature (Castel et al., 2002; Fransson et al., 2016). Those results addressed the influence of forest height and DBH on L-band SAR backscatter from ALOS-2 PALSAR-2 data. Other forest structure parameters derived from texture characteristics were proposed for sensitivity analysis of forest structure to SAR backscatter (Champion et al., 2013; Stelmaszczuk-Górska et al., 2018).

Although previous studies explored several effects of forest structure on SAR backscatter, the related studies should extend to the different forest and get deep and fully understanding of biomass accumulation related forest structure parameters. Given this need, this study explores the effect of mean forest height, mean DBH and canopy density on L-band SAR backscatter. The main objectives of this study are: (1) to investigate the difference of effects of the three biomass-accumulation related forest structure parameters on L-band SAR backscatter; and (2) to investigate the biomass-accumulation related forest structure parameters dependency of L-band backscatter. Here the biomass-accumulation related forest structure parameters include average canopy density, average forest height, and average forest DBH.

Materials and Methods

Study area

The study area is located in Xunke County, Heilongjiang Province in the northeast China (centered at 48°44′N, 128°20′E; Fig. 1). Its north lies Russia with a 14 kilometers frontier. The area belongs to the cold temperate continental monsoon climate and comprises approximately 11,274 km2 of managed and natural forests. The managed forests, covering approximately 80% of the forest area, include three dominant tree species: Oak (Xylosmaracemosum), Birch (Betulaplatyphylla Suk) and Aspen (Populusussuriensis). Most frequent tree species of the natural forests, contributing 20% of the forest area, are Larch (Abiesnephrolepis), Mongolian scotch pine (MongolicaLitv), and Pinuskoraiensis (PinuskoraiensisSieb). The elevation of the area ranges from 180 m to 560 m above sea level. The annual precipitation here varies from 600 mm to 800 mm. Most of it comes from rainfall in July and August. The annual mean temperature is 2.8  °C with a short summer of 130 frost-free days and a long winter of about 260 frozen days.

Figure 1 The location of the study area.

Data collection and data processsing

SAR data and its processing

SAR data collection.

ALOS-1, which was launched on January 24, 2006, was equipped with the PALSAR using a center frequency of 1279 MHz. The system enables image acquiring in single, dual, and quad polarization using the Stripmap mode and global coverage of spatial and temporal consistency. As the capability of its PALSAR image acquiring, it is possible to extract necessary parameters from L-band PALSAR images to monitoring global forest resources. In this study, the Stripmap mode data were acquired from an ascending orbit using a right-looking radar. These data, acquired at two adjacent orbit tracks, were supposed to cover the whole study area (Table 1). Three images at track 421–970 and another three at 421–980, acquired from June to September in 2007, were available in Fine Beam Dual-pol (FBD) mode at HH and HV polarization, with incidence angle of 38.7° and the original image pixel spacing was 9.5 m in slant range and 4.5 m in azimuth. All the data were level 1.1 product with single-look complex images.

Table 1 The details of acquired ALOS-1 PALSAR-1 data.

Track ID	Polarization	Imaging config	Acquisitiondate	
421-970	HH/HV	FBD	2007∕06∕22
2007∕08∕07
2007∕09∕22	
421-980	HH/HV	FBD	2007∕06∕22
2007∕08∕07
2007/09/22	

SAR data preprocessing.

SAR data preprocessing included radiometric calibration, multi-looking, gamma map filtering, and geo-referencing, whereby the radiometric calibration was done via the following equation: (1) σ0dB=10∗ logDNi2+KdB

where σ0 = backscattering coefficient, DNi = digital number of each pixel, KdB is the calibration coefficient with the value of −83.4 for FDB ALOS-1 PALSAR-1 images. Depending on the processing based on Eq. (1), the digital number of each pixel DNi was transformed into a backscattering coefficient sigma naught (σ0) in decibels (dB) (Englhart, Keuck & Siegert, 2011). Multi-looking of 5 both in range and azimuth direction were carried out to reduce noise. To further reduce random noise, we applied gamma mapper filter on the multi-looked images. The reason for using gamma mapper filter is that its performance shows clear noise reduction without substantial loss in resolution (Bernier et al., 2002). Geo-referencing was conducted in GAMMA 2009 software, using the digital elevation model (DEM) acquired by SRTM (Shuttle Radar Topography Mission) with 25 m posting.

Field data and its processing

Field data collection.

Field inventory map, which is completed every 5 to 10 years by each forest Bureau, is one of the products from the Chinese National Forest Inventory, named Forest Management Inventory (FMI). The map forest includes sub-compartment boundaries of each polygon. The polygons, called sub-compartment, are the basic inventory and mapping units of FMI. The sub-compartment boundaries are determined according to physiognomy of forest and generated from geographic information system (GIS) software where SPOT 5 (Systeme Probatoired Observation de la Terre) images and topographic maps service as a reference. The biophysical parameters we collected in the field inventory map include mean stand stem volume, mean forest height, mean forest age, mean DBH, sub-compartment canopy density and dominated tree species. The mean forest height, mean forest age and mean DBH are taken as the dimension of a standard tree. The volume of a standard tree is obtained from a tree volume table. The mean stand stem volume is calculated by the total volume of the sub-compartment and that of the standard tree.

Field data processing.

The goal of field data processing here was to improve the accuracy of inventory data and then reduced the uncertainties resulted from the field data application. The processing was carried out via the following operations. First, we removed sub-compartments with an area less than 2 hectare (ha.) to match the resolution of SAR image and DEM data. Second, 20 m buffers inside and along the selected stand boundaries were generated to minimize mis-registration effects for the inventory data and reduce overestimation caused by boundary uncertainties. After the operation, 6,037 sub-compartments were chosen from the original 10024. Details were reported in Table 2, respectively.

Table 2 The details of biophysical parameters from all sub-compartments and chosen in this study.

	All (10224)	Chosen (6037)	
Name	Max	Min	Mean	SD	Max	Min	Mean	SD	
Canopy density	1	0	0.56	0.14	0.90	0.20	0.54	0.16	
Mean forest age (year)	80	3	28.88	10.70	70	3	28.35	9.40	
Mean forest height (m)	20	1	9.40	2.90	17	2	9.61	3.01	
Mean DBH (cm)	36	1	10.00	3.85	36	1	9.82	3.70	
Mean stem volume (m3/ha)	198	0	48.74	27.99	198	2.0	47.62	30.91	
Sub-compartment area (ha)	184	0.35	21.61	16.82	184	2.3	24.336	19.58	

Ancillary data and its preprocessing

Ancillary data collection.

DEM data acquired by Shuttle Radar Topography Mission (SRTM) with InSAR (interferometric SAR) technique were collected for this study. According to the SRTM mission, SRTM data were anticipated to have an absolute horizontal circular accuracy of less than 20 m. Absolute and relative vertical accuracy was expected to be less than 16 and 10 m, respectively. However, the absolute vertical accuracy showed by the SRTM project team with performance estimations, is approximately 5 m. In our study area, DEM data were released at a spacing of 1 arc-second (∼30 m). Projection of the data was Universal Transverse Mercator (UTM) with coordinates of the World Geodetic System 1984 Coordinate System (WGS84). The data are accessible via the USGS (United States Geological Survey) seamless data by the United States Geological Survey (http://seamlesss.usgs.gov).

Ancillary data preprocessing.

In this study, SRTM DEM data were preprocessed for SAR images geocoding in GAMMA software. The procedure includes two steps: DEM parameter file generation and missing values correction. The first step of the procedure involved DEM geographic information input to generate a DEM parameter in GAMMA software. The input information in this step contains DEM title, data format, DEM height of offset, DEM width and so on. The second step involved retrieval of all values equal to −32,768 from original DEM data, replacing them with zero and interpolating the area with values equal to zero. The pixels with a value equal to −32,786 are also called missing values. They characterized by small gaps in DEM data.

Sensitivity analysis of SARσ0 to forest biophysical parameters

For each image, the sub-compartment-wise mean backscattering coefficients were calculated. The procedure was performed in ArcGIS 10.2 software via the function of Zonal Statistics. To better understand the signatures of the backscatter concerning each of forest structure related biophysical parameters and reduce effects resulted from environmental conditions, Subsets were considered here. The subsets were obtained by limiting one forest structure parameter as a constant. Especially, the backscatter coefficients from three acquired date at each sub-compartment were averaged firstly. Then the averaged backscatter coefficients were analyzed in relation to the following conditions: (1) radar backscatters change with mean forest height for a given canopy density; (2) radar backscatters vary with mean DHB for a given canopy density; (3) radar backscatters vary with canopy density for a given mean forest height; (4) radar backscatters change with mean DBH for a given mean forest height. Tree-species composition at the sub-compartment was available at the study area, however, the analysis including tree-species was not considered in this study, because all of the left 3 tree-species are coniferous species, which has insignificant tree-species dependence of the L-band SAR backscatter.

Sensitivities between the above-mentioned biophysical parameters and SAR backscatter were described by Pearson Correlation Coefficient (R), which is useful for linear correlation analysis between two variables (Ahlgren, 2003); Maximal Information Coefficient (MIC), which measures the dependence of two-variable relationship, both linear and non-liner (Reshef et al., 2011); the coefficient of determination (R2) of linear, Logarithmic and Quadratic, which describes the data relative to the regression function (Zhang et al., 2018).

Results

In our study, the HV-polarized band revealed higher sensitivity to all the forest structure related biophysical parameters than the HH-polarized band (Manabu et al., 2006; Santoro et al., 2009). Then it was chosen as a representative of the SAR signature here to analyze the effects of forest structure on SAR backscatter signatures. To investigate the SAR backscatter signatures for different forest structures, scatterplots were selected and used in showing the R performance. Figures 2–5 show the linear correlations of each forest structural parameter with HV backscatter. During the analysis, 6,037 samples were divided into distinct groups depending on mean canopy density and mean forest height. The details for sample grouping based on mean canopy density were shown in Tables 3 and 4 as representatives because of their better performance. The results we explored were demonstrated in the following 4 parts according to given constant forest biophysical parameters.

Figure 2 The scatter plots between forest height and HV backscatter at given canopy densities.

The seven subfigures described the tendency charts of HV backscatter changing with Mean forest height at canopy density equaling to 0.2 (A), 0.3 (B), 0.4 (C), 0.5 (D), 0.6 (E), 0.7 (F), and 0.8 (G), respectively.

Figure 3 The scatter plots between mean forest DBH and HV backscatter at given canopy densities.

The seven subfigures described the tendency charts of HV backscatter changing with Mean DBH at canopy density equaling to 0.2 (A), 0.3 (B), 0.4 (C), 0.5 (D), 0.6 (E), 0.7 (F), and 0.8 (G), respectively.

Figure 4 The scatter plots between canopy density and HV backscatter at given mean forest height.

Mean forest height was divided into four groups, in the first group, mean forest height ranges from 2 m to 5 m (A), in the second group, mean forest height ranges from 6 m to 9 m (B), in the third group, mean forest height ranges from 10 m to 13 m (C), in the fourth group, mean forest height ranges from 14 m to 17 m (D).

Figure 5 The scatter plots between mean DBH and HV backscatter at given mean forest height.

Mean forest height was divided into four groups, in the first group, mean forest height ranges from 2 m to 5 m (A), in the second group, mean forest height ranges from 6 m to 9 m (B), in the third group, mean forest height ranges from 10 m to 13 m (C), in the fourth group, mean forest height ranges from 14 m to 17 m (D).

Table 3 The correlations between HV backscatter coefficients and mean forest height at different canopy density.

Canopy
Density	Correlation	Samples	R2	
	Linear (R)	Non-linear (MIC)		Linear	Logarithmic	Quadratic	
0.2	0.72	0.42	189	0.53	0.42	0.60	
0.3	0.75	0.72	617	0.56	0.37	0.77	
0.4	0.85	0.72	320	0.73	0.53	0.89	
0.5	0.86	0.55	1215	0.73	0.52	0.86	
0.6	0.98	0.72	1885	0.96	0.91	0.96	
0.7	0.86	0.72	1235	0.75	0.56	0.90	
0.8	0.92	0.72	646	0.84	0.89	0.86	

Table 4 The correlation coefficients HV between backscatter coefficients and DBH at different canopy density.

Canopy
Density	Correlation	Samples	R2	
	Linear (R)	Non-linear (MIC)		Linear	Logarithmic	Quadratic	
0.2	0.19	0.16	189	0.04	0.05	0.12	
0.3	0.00	0.32	617	0.00	0.01	0.07	
0.4	0.50	0.25	320	0.25	0.23	0.25	
0.5	0.32	0.53	1215	0.10	0.10	0.11	
0.6	0.09	0.75	1885	0.01	0.06	0.23	
0.7	0.41	0.76	1235	0.17	0.23	0.39	
0.8	0.05	0.42	646	0.00	0.03	0.26	

Forest height Sensitivity to L-band SAR backscatter coefficients with given canopy density

Figure 2 showed the behaviors of HV backscatter coefficients for different mean forest height at a given mean canopy density in a linear correlation performance. The HV backscatter coefficients in each scatterplot in Fig. 2 were found to increase from low forest height to high forest height. At the same mean forest height, the level of the backscatters at low mean canopy density area was higher than that at a high mean canopy density area. However, it was not the case for mean canopy density equal to 0.2 and 0.8. The associations between forest height and HV backscatter were fluctuant, but they indicated a high correlation between them especially when the values of mean canopy density were higher than 0.3. The highest R between forest height and HV backscatter was obtained at mean canopy density equal to 0.6 and the value of R was 0.97.

Table 3 gathered detail information of divided groups relying on mean canopy density, the averaged samples relying on values of mean forest height and R, MIC and R2 values between HV backscatter coefficients and mean forest height. The results in Table 3 demonstrated great effects of mean canopy density on the relationship between HV backscatter and mean forest height. The better R values between them correlated well when mean canopy densities greater than 0.4 and were 0.71 for canopy density equal to 0.4, 0.76 for 0.5, 0.97 for 0.6, 0.70 for 0.7 and 0.82 for 0.8, respectively. Moderate correlations were obtained when mean canopy density lower than 0.4. The values of MIC are around at 0.72 except canopy density equals 0.2 or 0.5. From Table 3, We can see both linear and several non-linear functions have better performance for mean forest height retrieval when both of the values of R and MIC are higher than 0.7. The R2 values of these retrieval functions are higher in these functions. The results also revealed the obvious effects of mean forest height and mean canopy density on SAR backscatter.

DBH sesitivity to L-band SAR backscatter coefficients with given canopy density

The results shown in Fig. 3 and Table 4 are for the effects of DBH at given canopy density. To get an understanding of DBH effects on HV backscatter coefficients, sets of scatterplots depending on different mean canopy density were analyzed. To simplify the analysis procedure and give a clear tendency, sub-compartment samples were averaged according to mean DBH. The details of the simplification were presented in Table 4.

There were no obvious effects of mean canopy density on the relationship between HV backscatter coefficients and mean DBH. HV backscatter coefficients fluctuate with the variation of mean DBH. According to Table 4, the linear correlations between HV backscatter and mean DBH are lower than mean forest height. The non-linear correlations are higher than linear correlations, but the MIC values higher than 0.7 were gotten at canopy density equal 0.6 or 0.7. While the highest R value is acquired at 0.7. The lower values of R and MIC in Table 4 revealed the low sensitivity of DBH to HV backscatter signatures. They also suggested low effects of mean canopy density on the relationship of HV backscatter coefficients and mean DBH. Nonetheless, we should pay more attention to the different dynamic range of mean DBH at different mean canopy density, the results showed weak connection appeared when the mean DBH dynamic range was small, here the dynamic ranges from 0 to 50 cm. but a better correlation with larger DBH dynamic range, like the range between 5 to 85 cm, 10 to 90 cm and 5 to 115 cm.

Canopy density sesitivity to L-band SAR backscatter coefficients with given mean forest height

HV backscatter coefficient dependence on canopy density was investigated at a constant mean forest height. We divided all the mean forest height into 4 groups to analyze the HV backscatter coefficient dependence on canopy density. The height interval of each group was 4 m because observing errors of mean sub-compartment height was around 3 or 4 m. In the first group, the forest height varies from 2 to 5 m, the second from 6 to 9 m, the third from 10 to 13 m and the last group from 14 to 17 m. The results indicated obvious canopy density dependence of HV backscatter coefficients especially when mean forest height higher than 5 m. Figure 4 and information at up part of the Table 5 recorded the details of the interactions between HV backscatter coefficients and mean canopy density. According to Fig. 4 and Table 5, mean forest height only showed less obvious effects on the relationship between HV backscatter coefficients and mean canopy density, especially when the mean forest height lower than 5 m. The R and MIC values between HV backscatter coefficients and mean canopy density were similar when mean forest height ranges from 6 to 9 m and 10 to 13 m. it revealed that mean canopy density has great sensitivities to HV backscatter coefficients. HV backscatter coefficients increased with the increasing of mean canopy density when mean forest height was higher than 5 m. All of the values of R between them were higher than 0.70 and the highest R was 0.95.

Table 5 The correlation coefficients between HV backscatter coefficients and DBH at different canopy density.

Mean Forest Height (m)	Correlation	R2	
		Linear (R)	Non-linear (MIC)	Linear	Logarithmic	Quadratic	
Canopy density	2–5	0.03	0.52	0.00	0.01	0.55	
6–9	0.97	0.99	0.95	0.91	0.95	
10–13	0.97	0.99	0.95	0.93	0.96	
14–17	0.72	0.47	0.52	0.45	0.59	
DBH	2–5	0.27	0.52	0.07	0.05	0.74	
6–9	0.03	0.47	0.00	0.00	0.58	
10–13	0.20	0.47	0.04	0.04	0.06	
14–17	0.61	0.99	0.37	0.37	0.44	

DBH sesitivity to L-band SAR backscatter coefficients with given mean forest height

The plot of Fig. 5 showed the DBH effects on HV backscatter at a given mean forest height. The way of DBH sensitivity analysis was in line with canopy density analyzed in 3.3. All of the selected samples were divided into 4 groups according to forest height and then HV backscatter coefficient values were averaged with the same DBH value. R, R2 and MIC values between HV backscatter coefficients and DBH were presented in the bottom part of the Table 5. These lower values revealed the insensitiveness of DBH to HV backscatter coefficients. All of them were less than 0.1 except the group with mean forest height ranging from 14 m to 17 m.

Forest Species Sensitivity to L-band SAR backscatter coefficients

Three dominated forest species including Oak (Xylosmaracemosum), Birch (Betulaplatyphylla Suk) and Aspen (Populusussuriensis) were selected to analyze forest species effects on L band HV backscatter. All of them were divide into 6 groups according to mean forest canopy density firstly, then the correlations and relationships between mean forest age and SAR backscatter, mean forest height and SAR backscatter, and mean DBH and SAR backscatter are analyzed, respectively. The correlation between mean forest height and HV backscatter in Birch (Betulaplatyphylla Suk) and Aspen (Populusussuriensis) have similar trend with that analyzed without distinguishing forest species. However, the correlation between mean DBH and HV backscatter in three forest species show obvious increase even their values show fluctuation with varying of mean canopy density. We also analyzed the forest age effects on HV SAR backscatter for each dominated forest species. Forest age show lower sensitivity than mean forest height and mean DBH, it shows almost no difference in three species.

Oak (Xylosmaracemosum) forest structural parameters effects on L-band HV SAR backscatter

In Table 6 we described forest age effects on L-band HV SAR backscatter. R and MIC values were used to analyze the correlation between mean forest age and backscatter, R2 applied here to describe the regression function performance. Best performance is at mean canopy density equals 0.5 where R = 0.90 and MIC = 1. The worst performance appears when canopy density equals 0.2. Table 7 shows the information of mean DBH affecting on backscatter. When canopy density equals 0.6, mean DBH show the highest correlation with SAR backscatter with R = 0.85 and MIC =0.99. Table 8 shows the sensitivity of mean forest height to SAR backscatter. It only shows higher correlation when canopy density between 0.5 and 0.6.

Table 6 Oak (Xylosmaracemosum) forest age effects on L band HV SAR backscatter.

Canopy
Density	Correlation	R2	
	Linear (R)	Non-linear (MIC)	Linear	Logarithmic	Quadratic	
0.2	0.15	0.20	0.02	0.01	0.18	
0.3	0.68	0.52	0.47	0.61	0.64	
0.4	0.51	0.46	0.26	0.08	0.72	
0.5	0.90	1	0.81	0.73	0.84	
0.6	0.78	1	0.61	0.64	0.66	
0.7	0.62	0.55	0.39	0.32	0.44	

Table 7 Oak (Xylosmaracemosum) forest DBH effects on L band HV SAR backscatter.

Canopy
Density	Correlation	R2	
	Linear (R)	Non-linear (MIC)	Linear	Logarithmic	Quadratic	
0.2	0.27	0.22	0.07	0.12	0.37	
0.3	0.58	0.47	0.33	0.58	0.88	
0.4	0.72	0.99	0.52	0.29	0.53	
0.5	0.51	0.68	0.26	0.30	0.39	
0.6	0.85	0.98	0.73	0.79	0.85	
0.7	0.48	0.29	0.23	0.10	0.53	

Table 8 Oak (Xylosmaracemosum) forest height effects on L band HV SAR backscatter.

Canopy
Density	Correlation	R2	
	Linear (R)	Non-linear (MIC)	Linear	Logarithmic	Quadratic	
0.2	0.4	0.378879	0.16	0.1387	0.1688	
0.3	0.11045361	0.469565	0.0122	0.0803	0.6669	
0.4	0.486826458	0.311278	0.237	0.1252	0.4736	
0.5	0.911153116	0.99403	0.8302	0.7842	0.7842	
0.6	0.814555093	1	0.6635	0.7005	0.7491	
0.7	0.313687743	0.311278	0.0984	0.0842	0.1527	

Birch (Betulaplatyphylla Suk) forest structural parameters effects on L-band HV SAR backscatter

Tables 9–11 show the effects of Birch (Betulaplatyphylla Suk) forest structure on L band SAR backscatter. They described forest age effects, forest DBH effects and forest height effects, respectively. Forest age, forest DBH and forest height show obvious correlation with HV backscatter when mean canopy density greater than 0.5. Linear correlation is more obvious than non-linear correlation between these forest structural parameters and backscatter.

Table 9 Birch (Betulaplatyphylla Suk) forest age effects on L band HV SAR backscatter.

Canopy
Density	Correlation	R2	
	Linear (R)	Non-linear (MIC)	Linear	Logarithmic	Quadratic	
0.2	−0.10	0.14	0.01	0.03	0.01	
0.3	−0.16	0.165	0.02	0.00	0.38	
0.4	0.46	0.59	0.22	0.14	0.28	
0.5	0.66	0.55	0.43	0.46	0.44	
0.6	0.71	0.32	0.50	0.59	0.61	
0.7	0.94	0.99	0.89	0.86	0.89	

Table 10 Birch (Betulaplatyphylla Suk) forest DBH effects on L band HV SAR backscatter.

CanopyDensity	Correlation	R2	
	Linear (R)	Non-linear (MIC)	Linear	Logarithmic	Quadratic	
0.2	0.04	0.29	0.00	0.00	0.01	
0.3	0.46	0.30	0.21	0.21	0.21	
0.4	0.79	0.46	0.62	0.59	0.62	
0.5	0.70	0.47	0.49	0.70	0.75	
0.6	0.81	0.36	0.65	0.84	0.81	
0.7	0.90	1	0.80	0.76	0.76	

Table 11 Birch (Betulaplatyphylla Suk) forest height effects on L band HV SAR backscatter.

Canopy
Density	Correlation	R2	
	Linear (R)	Non-linear (MIC)	Linear	Logarithmic	Quadratic	
0.2	0.31	0.36	0.12	0.09	0.15	
0.3	0.61	0.15	0.37	0.31	0.46	
0.4	0.83	0.51	0.69	0.57	0.73	
0.5	0.83	0.46	0.70	0.76	0.73	
0.6	0.86	0.50	0.74	0.81	0.80	
0.7	0.65	0.65	0.43	0.34	0.57	

Aspen (Populusussuriensis) forest structural parameters effects on L-band HV SAR backscatter

The correlations of Aspen (Populusussuriensis) forest structural parameters and L band HV SAR backscatter are depicted in Tables 12–14. In three forest structural parameters, forest height shows the best linear correction with HV backscatter with all of the R values greater than 0.4. However, all of the effects of three forest structural parameters fluctuated with the variation of mean canopy density.

Table 12 Aspen (Populusussuriensis) forest age effects on L band HV SAR backscatter.

CanopyDensity	Correlation	R2	
	Linear (R)	Non-linear (MIC)	Linear	Logarithmic	Quadratic	
0.2	0.42	0.42	0.17	0.02	0.62	
0.3	−0.03	0.31	0.00	0.02	0.24	
0.4	0.75	0.41	0.56	0.46	0.62	
0.5	0.47	0.64	0.22	0.24	0.27	
0.6	0.51	0.65	0.26	0.13	0.30	
0.7	0.77	0.61	0.57	0.68	0.68	

Table 13 Aspen (Populusussuriensis) forest DBH effects on L band HV SAR backscatter.

CanopyDensity	Correlation	R2	
	Linear (R)	Non-linear (MIC)	Linear	Logarithmic	Quadratic	
0.2	0.45	0.40	0.20	0.03	0.34	
0.3	−0.08	0.40	0.01	0.07	0.21	
0.4	0.80	0.69	0.65	0.52	0.65	
0.5	0.18	0.59	0.03	0.06	0.11	
0.6	0.73	0.74	0.54	0.53	0.60	
0.7	0.68	0.65	0.46	0.55	0.60	

Table 14 Aspen (Populusussuriensis) forest height effects on L band HV SAR backscatter.

CanopyDensity	Correlation	R2	
	Linear (R)	Non-linear (MIC)	Linear	Logarithmic	Quadratic	
0.2	0.69	0.65	0.47	0.35	0.57	
0.3	−0.54	0.20	0.29	0.26	0.36	
0.4	0.71	0.68	0.51	0.35	0.79	
0.5	0.45	0.38	0.20	0.20	0.24	
0.6	0.87	0.71	0.76	0.61	0.83	
0.7	0.82	0.71	0.67	0.52	0.80	

Discussion

The objective of this study is to explore the effects of forest structure related biophysical parameters on L-band SAR backscatter. We demonstrated different sensitivity of mean canopy density, mean forest height and mean DBH via a given interest factor as a constant. 6037 sub-compartment samples were involved in our study to analyze forest biomass accumulation related forest structure effects on L-band SAR backscatter. Scatter plots and correlation analysis were used in this study. Based on this method, our findings revealed that: (1) mean canopy density is more sensitive to HV backscatter, as well as mean forest height. However, the effect of mean DBH on HV backscatter is unobvious; (2) the forest structure dependencies of L-band SAR backscatter varies with the level of mean canopy density and mean forest height and dynamic ranges of DBH. Forest structure effects on L-band SAR backscatter are obvious, however, those effects changed with different parameters. One of the key results of our research was that L-band SAR backscatter showed a strong dependency on mean canopy density and mean forest height. Other studies in the boreal forest showed the similar influence of mean canopy density to SAR coherence, but in these studies, canopy density is defined as area fill factor (Cartus, 2010; Nguyen et al., 2016). More recently, Luong and Manabu argued high sensitivity of L-band SAR backscatter to tree height, which also agreed with our results (Matsuoka et al., 2006; Nguyen et al., 2016). The results of Askne et al. demonstrated the effects of mean forest height and canopy density in a different way that is based on the scattering mechanism of the forest. In their researches, they assumed that SAR interferometric coherence is mainly determined by forest scattering height if the scattering components coming from land surface can be neglected. If the ground contribution is obvious, canopy density shows more influence on SAR interferometric coherence. They used a different method to quantify the influence of canopy density and forest height, however, their results also confirmed the sensitivity of L-band coherence to canopy density and forest height (Askne et al., 1997). The results from Luong and Matsuoka also revealed dependency of L-band SAR backscatter on basal area, in the research of Luong, R2 between them are 0.48 and 0.28 in a dry season and a rainy season, respectively. The unobvious correlation between DBH and L-band SAR backscatter in our study could be explained by the fact that the DBH value interval in our study was small, all of them were less than four cm. However, in the research of Manabu, the value interval is 10 cm (Manabu et al., 2006).

Forest structure effects analysis based on forest microwave scattering models at different bands and with different parameters also confirmed forest structure dependency of SAR backscatters. It means that there is a continued need for a suitable way for forest structure analysis before AGB estimation with SAR backscatter. For example, Ni confirmed the heterogeneous influence caused by forest height effect on forest biomass inversion with simulated data. After the correction of forest height effects on L-band SAR backscatter, R2 between the truth data and simulated results are higher than 0.8 (Ni et al., 2013). Forest structure influence based on Random vegetation over the ground (RVoG) model showed the physical reason and practical way to detect SAR backscatter sensitivity to forest structure (Cloude & Papathanassiou, 2003; Cloude, 2006; Solberg et al., 2013). Since the spatial heterogeneity of forest structure over landscapes, it required a simple and effective method for forest structure influence when AGB was estimated with SAR backscatter. The method used in this research demonstrated a simple and effective way for the required analysis.

This study confirmed the effectiveness of using stand samples to analysis the forest structure sensitivity to SAR backscatter. It makes sure the influence of forest canopy and forest height on L-band SAR backscatter. It also interpreted the way how forest structure affects backscatter in the test sites. Despite forest structure attributes varies all over the Earth, even the method applied in this study can be applied in other sites, the quantification of forest structure to different SAR data needs to further analyzed according to their unique characters.

Conclusions

Following previous research, in this paper, HV channel backscatter was chosen as a representative of the L-band SAR signature to explore forest structure diversity to them. L-band SAR backscatter dependence on canopy density, mean forest height, mean DBH and species of forest sub-compartment were explored in this study. Based on the analysis of the 6037 forest sub-compartment in northeast China, we concluded that canopy density showed the greatest influence on L-band SAR backscatter with all R higher than 0.7. Forest structure parameter following is mean forest height, however, its impact on L-band SAR backscatter depends on sub-compartment canopy density. It showed a lower sensitivity when canopy density was lower, while it showed higher sensitivity when canopy density was higher. Sensitivities of DBH to backscatter are unremarkable in this study. However, the sensitivity of L-band SAR backscatters grows with the increase of DBH dynamic intervals, especially with higher values of DBH. Forest species also affect the sensitivity of forest mean canopy density, canopy forest height and forest DBH to L-band SAR backscatter. Their influence depends on different tree species. This study confirmed the influence of forest structure to SAR backscatter at the forest sub-compartment level. It also quantifies forest structure dependence of L-band SAR backscatter. An obvious limitation of this study is not considering the detail influence of tree species, especially according to the analysis with given the mean forest height as a constant. Because of the limited stand samples in different mean forest height groups, its influence needs further explored in the future. Moreover, the relationship between canopy density, forest height and DBH, and forest AGB should be analyzed in the future. And then forest structure dependence of L-band SAR backscatter can be further explored with the distinction of more tree species. It would extend the application capability of this study.

Supplemental Information

Supplemental Information 1 Raw data

Click here for additional data file.

We would like to thank XunkeForestry Department for its assistance and interpretation of ground data. We are also grateful for the technology and SAR data support from the research group of Zengyuan Li in the Institute of Forest Resource Information Techniques, Chinese Academy of Forestry.

Additional Information and Declarations

Competing Interests

Author Contributions

Data Availability

The authors declare there are no competing interests.

Yongjie Ji conceived and designed the experiments, performed the experiments, analyzed the data, prepared figures and/or tables, authored or reviewed drafts of the paper, and approved the final draft.

Jimao Huang, Yilin Ju and Shipeng Guo performed the experiments, prepared figures and/or tables, and approved the final draft.

Cairong Yue analyzed the data, authored or reviewed drafts of the paper, and approved the final draft.

The following information was supplied regarding data availability:

Raw data are available as Supplementary Files.

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
