# Peer review of "Forest structure dependency analysis of L-band SAR backscatter"

_PeerJ, doi:10.7717/peerj.10055_

## Round 0.1 · original submission · Major Revisions

Currently, we have collected 2 reports, both request that you make major revisions. Please revise your manuscript accordingly.

Reviewer 1 ·

Basic reporting

please see the attachments.

Experimental design

please see the attachments.

Validity of the findings

please see the attachments.

Additional comments

please see the attachments.

Annotated reviews are not available for download in order to protect the identity of reviewers who chose to remain anonymous.

Reviewer 2 ·

Basic reporting

This study examined L-band SAR backscatter sensitivity to canopy density, tree height and DBH at the sub-compartment level and concluded that canopy density showed a great influence on L-band backscatter than mean forest height and DBH and the sensitivity of L-band backscatter to tree height and DBH was affected by canopy density. The manuscript is well written and presented.

Experimental design

Regarding the metholodogy, there are still some major concerns that should be addressed.

1. The authors used Pearson Correlation Coefficient R to analyze the sensitivities between forest parameters (i.e., canopy density, tree height and DBH) and radar backscatter. As shown in Figures 2-5, the Pearson Correlation Coefficient seems to be linear model. It is concerned that the linear model may not be appropriate for this study because more advanced nonlinear models may be commonly used for retrieving forest parameters from radar images. When using the different models, the retrieved relationship between forest parameters and radar backscatter could be very different. For example, forest height may not have a strong correction with radar backscatter when a linear model is used for the investigation, but they may be highly related when turning to a nonlinear model. Therefore, I suggest the authors to conduct a more extensive literature review on the models that are commonly used for radar remote sensing of forest parameters and used these models instead of the linear model to examine the sensitivities between forest parameters and radar backscatter. Otherwise, the findings and conclusions of this study may not be convincing.

2. Are there mountain areas in the study area? It is well known that radar backscatter is affected by the terrain. For example, forests in the slope facing the SAR sensor feature stronger radar backscatter than forests in the shadow or flatten areas. How did this study deal with such terrain effect?

3. Since the SAR images were acquired at the different dates in the period from June to September, the SAR backscatter of forest may be affected by forest phenology. Did the authors consider such phenology effect in this study?

Validity of the findings

no comment

Additional comments

no comment

---

## Round 0.2 · accepted · Accept

After reading the manuscript and your response letter, I agree with the recommendations of the two reviewers and am therefore happy to formally accept your paper for publication in PeerJ. I would like to congratulate you on your work and thank you for considering PeerJ for its publication.

Reviewer 1 ·

Basic reporting

no comment.

Experimental design

no comment.

Validity of the findings

no comment.

Additional comments

In this new manuscript, all of the issues I mentioned have been modified, only one problem. The expressions and your reasoning are now clearer and profounder.
The authors have added the new experiment to show the correlations and relationships between mean forest age and SAR backscatter, mean forest height and SAR backscatter, and mean DBH and SAR backscatter, respectively, which are listed in Table 6-14. I think it’s too lengthy. Under this condition, I suggest these 9 tables should be merged into one table for brevity.

Reviewer 2 ·

Basic reporting

no comment

Experimental design

no comment

Validity of the findings

no comment

Additional comments

The authors have addressed all my concerns. I have no further comments.